# Fabrication of Low-Molecular-Weight Hyaluronic Acid–Carboxymethyl Cellulose Hybrid to Promote Bone Growth in Guided Bone Regeneration Surgery: An Animal Study

**DOI:** 10.3390/polym14153211

**Published:** 2022-08-06

**Authors:** Chun-Yu Lin, Po-Jan Kuo, Ya-Hui Lin, Chi-Yu Lin, Jerry Chin-Yi Lin, Hsien-Chung Chiu, Tsung-Fu Hung, Hung-Yun Lin, Haw-Ming Huang

**Affiliations:** 1Department of Medical Imaging, Taipei Medical University Hospital, Taipei 11031, Taiwan; 2Department of Periodontology, Tri-Service General Hospital, Taipei 11490, Taiwan; 3School of Dentistry, National Defense Medical Center, Taipei 11490, Taiwan; 4School of Dentistry, College of Oral Medicine, Taipei Medical University, Taipei 11031, Taiwan; 5Graduate Institute of Cancer Molecular Biology and Drug Discovery, College of Medical Science and Technology, Taipei Medical University, Taipei 11031, Taiwan

**Keywords:** low-molecular hyaluronic acid, bone regeneration, MAPK pathways, grafting material

## Abstract

Guided bone regeneration surgery is an important dental operation used to regenerate enough bone to successfully heal dental implants. When this technique is performed on maxilla sinuses, hyaluronic acid (HLA) can be used as an auxiliary material to improve the graft material handling properties. Recent studies have indicated that low-molecular hyaluronic acid (L-HLA) provides a better regeneration ability than high-molecular-weight (H-HLA) analogues. The aim of this study was to fabricate an L-HLA-carboxymethyl cellulose (CMC) hybrid to promote bone regeneration while maintaining viscosity. The proliferation effect of fabricated L-HLA was tested using dental pulp stem cells (DPSCs). The mitogen-activated protein kinase (MAPK) pathway was examined using cells cultured with L-HLA combined with extracellular-signal-regulated kinase (ERK), c-Jun N-terminal kinase (JNK), and p38 inhibitors. The bone growth promotion of fabricated L-HLA/CMC hybrids was tested using an animal model. Micro-computer tomography (Micro-CT) and histological images were evaluated quantitatively to compare the differences in the osteogenesis between the H-HLA and L-HLA. Our results show that the fabricated L-HLA can bind to CD44 on the DPSC cell membranes and affect MAPK pathways, resulting in a prompt proliferation rate increase. Micro CT images show that new bone formation in rabbit calvaria defects treated with L-HLA/CMC was almost two times higher than in defects filled with H-HLA/CMC (*p* < 0.05) at 4 weeks, a trend that remained at 8 weeks and was confirmed by HE-stained images. According to these findings, it is reasonable to conclude that L-HLA provides better bone healing than H-HLA, and that the L-HLA/CMC fabricated in this study is a potential candidate for improving bone healing efficiency when a guided bone regeneration surgery was performed.

## 1. Introduction

Missing teeth due to periodontal disease or dental caries is a problem commonly encountered in dentistry. According to the injury and amount of time the teeth are missing, bone defects can vary from minor to as much as a 60% reduction in the original bone [1]. Large bone defects often cause difficulties for dentists attempting to place dental implants in a suitable position. These defects are approached using guided bone regeneration surgery, a technique that regenerates bone mass and increases bone width to help artificial dental implants heal.

Although in certain cases bone tissue has the ability to heal on its own, defects above a certain size can exceed the bone’s capacity to heal by itself, making bone grafts necessary to guide and promote healing. Clinical studies have found that bone graft implantation can increase implant healing success rates from 81 to 95.6% [2]. Traditional commercial bone grafts can be divided into four categories: autografts (grafts harvested from the patient), allografts (grafts harvested from another person), xenografts (grafts derived from a different species), and alloplasts (artificial synthetic material) [3]. The major problem when using allografts and xenografts is immune rejection. Although autografts can reduce unnecessary transplant rejection [4], insufficient source material and the need for secondary surgery are issues that commonly arise. To overcome these problems, several osteoconductive and osteoinductive artificial materials have been developed for this surgery over the years. Among these materials, the most commonly used is a mixture of hydroxyapatite (HA) and β-calcium triphosphate (β-TCP) [5,6]. To increase bone growth efficiency, bio-complex hybrids with HA/β-TCP and biopolymers have been recently developed [7]. In addition, bone grafts such as a mixture of HA/β-TCP are difficult to handle at the posterior maxilla due to filling material sagging. Therefore, a matrix carrier is necessary to prevent a loss of bone graft material during operations. 

Hyaluronic acid (HLA) is a polysaccharide composed of D-glucuronic acid and N-acetylglucosamine [8], and a component of the extracellular matrix that exhibits excellent biocompatibility when applied in the human body [9]. For medical applications to increase bone growth efficiency, HLA has been used as a material to coat the surface of titanium dental implants [8]. Several experiments have provided evidence that HLA can also promote bone cell migration and growth [10,11]. Recent studies have attempted to combine HLA and other bone substitutes such as gelatin [12] and collagen [13] to form substitute bone composites. HLA has been reported as an effective matrix carrier to prevent ceramic filler from sagging during sinus augmentation [12,14]. However, the effect of HLA on bone quality remains controversial for bone grafts combined with HLA for alveolar socket preservation. A report from Elkarargy (2013) indicated that HLA can increase osteoconduction efficiency [15], though other investigations have demonstrated that linear hyaluronic acid did not result in bone defect healing [16]. Since the role of HLA in the bone graft system was to act as a structural enforcement material, high-molecular-weight (MW) HLA (H-HLA) has been used in the majority of previous systems for its high viscosity [17]. This high viscosity makes H-HLA’s osteoinductive properties unclear [18,19]. 

Unlike H-HLA’s unclear effects, low-MW HLA (L-HLA) (<10^3^ kDa) is well-known to provide a positive effect on cell proliferation and differentiation [19,20,21]. For example, HLA with an MW between 80-800 kDa has been reported to increase wound healing efficiency [22,23]. The positive effects of L-HLA on bone growth have also been recognized [20,24,25]. However, when the MW of HLA is reduced, handling becomes difficult due to a sharply reduced viscosity [21,25]. Considering this difficulty, Huang et al. (2019) created an L-HLA and carboxymethyl cellulose (CMC) hybrid and successfully used the new material to increase wound healing efficiency [23]. Interestingly, CMC has also been reported to exhibit a bone tissue engineering bio-function [26]. Although biopolymer composites combined with CMC and H-HLA have been developed for use as a tissue barrier [27] and nerve-repair material [28], no reports have discussed the bone growth-promoting effect of HLA/CMC biocomposite previously. For dental research, Huang et al. fabricated a novel hydrogel with L-HLA and CMC. Their study found this biocomposite to exhibit excellent barrier membrane properties and usefulness as a system to release anti-inflammatory drugs [29]. Although several in vivo and in vitro studies have investigated possible applications of L-HLA in osteo-regeneration, the effect of an L-HLA/CMC bone graft material on bone defect healing remains unclear. To address this knowledge gap, an L-HLA/CMC hybrid material was fabricated and its performance in treating bone defects was tested in an animal model.

## 2. Materials and Methods

### 2.1. Preparation of Low-Molecular-Weight Hyaluronic Acid

HLA with an MW of 3000 kDa (Cheng-Yi Chemical Industry Co., Ltd., Taipei, Taiwan) was used in this study. As in a previous study, the L-HLA used in this study was fabricated using γ-irradiation [23]. Briefly, HA powder was irradiated using a cobalt-60 irradiator (Point Source, AECL, IR-79, MDS Nordion International Co., Ltd., Ottawa, ON, Canada) at 22 °C for 20 h. The γ-irradiation dose at the sample location was set at 1 kGy/h, confirmed using alanine pellet dosimeters (FWT-50, Far West Technology, Inc., Goleta, CA, USA). The MW of γ-irradiated L-HLA was 200 kDa, as measured by gel permeation chromatography (Series 200, Perkin Elmer, Waltham, MA, USA), which has been described previously [23]. In this study, the unexposed sample was used as H-HLA.

### 2.2. Isolation and Culture of Dental Pulp Stem Cells

Dental pulp stem cells (DPSCs) were isolated from the wisdom molars of healthy donors which were extracted due to orthodontic treatment. All procedures were approved by TMU-Joint Institutional Review Board, Taipei Medical University, Taipei Taiwan (TMU-JIRB no. 201503064). The outgrowth method was used to isolate DPSCs according to previous studies [30,31]. Immediately after the teeth were extracted, pulp tissue was removed from the teeth and sliced into small pieces. These tissues were then cultured in 3.5 cm diameter Petri dishes at 37 °C in a 5% CO_2_ environment. The culture medium was α-minimal essential medium (α-MEM, Gibco/Invitrogen, Carlsbad, CA, USA) supplemented with 15% FBS, 1% antibiotic-antimycotic (Gibco/Invitrogen), and 100 µmol/L of L-ascorbic acid 2-phosphate (Sigma-Aldrich, St. Louis, MO, USA). After 70 to 80% confluence, the isolated dental pulp stem cells were passed through a 70 µm strainer (BD Falcon, San Jose, CA, USA).

### 2.3. Characteristics Identifying Dental Pulp Stem Cells

To characterize the dental pulp stem cells obtained, cells were identified using cell-surface markers. DPSCs were seeded in 10 mm dishes for 24 h and detached with trypsin–EDTA. Then, the cells were harvested and suspended in culture medium at a concentration of 1 × 10^6^ cells/mL. After centrifuging at 1000 rpm for 5 min, cells were incubated in the dark with fluorescein isothiocyanate (FITC)-conjugated antibodies in PBS at 4 °C for 30 min to label CD14, CD44 (AbD Serotech, Raleigh, NC, USA), CD34, STRO-1 (Santa Cruz Biotechnology, Santa Cruz, CA, USA), CD29 (Exbio, Praha, Czech Republic), CD73 (BD, Biosciences, Heidelberg, Germany), CD90, and CD105 (Biolegend, San Diego, CA, USA). Labeled DPSCs were washed twice with PBS and fixed in 4% paraformaldehyde for 10 min at 4 °C. Markers were detected using flow cytometry (Guava EasyCyte Mini Base System, Guava Technologies, Millipore, Hayward, CA, USA) and analyzing software (FlowJo software, TreeStar Inc., Ashland, OR, USA). Debris and dead cells were gated out using forward-side scatter plot. 

The isolated DPSCs’ ability to generate bone was tested using osteogenic induction. Before the experiment, 500 µL of 2 × 10^4^ cells/mL were cultured with an osteogenesis induction medium containing 1.8 mmol/L KH2PO4 and 0.01 µmol/L dexamethasone (Sigma-Aldrich, St. Louis, MO, USA) in 24-well plates. The medium was changed twice per week. After one month, cells were gently washed three times with PBS solution, fixed with 4% paraformaldehyde for 10 min, and stained with 250 mL of 2% Alizarin Red S solution (Sigma-Aldrich). After discarding the staining solution, cells were gently washed with PBS three times. Images of the stained cells were observed with a bright-field microscope (Eclipse TS100; Nikon Corporation, Tokyo, Japan) that incorporated a digital camera (SPOT Idea, Diagnostic Instruments Inc., Sterling Heights, MI, USA).

### 2.4. In Vitro Cellular Assessment

#### 2.4.1. The Effect of L-HLA on DPSC Viability

To determine effective L-HLA concentration, cell proliferation assessments were performed. DPSCs (2 × 10^4^ cells/mL) were co-cultured with prepared L-HLA at concentrations of 0, 0.5, 1, 2, and 4 mg/mL. All experiments were performed FBS-free. After culturing for three days, MTT solution (3-(4,5-dimenthylthiazol-2-yl)-2,5-diphenyltetrasolium bromide) (Roche Applied Science, Mannheim, Germany) was added to the culture well and incubated for 4 h. Then, DMSO was added to the sample to dissolve formazan crystals. Formazan crystals, whose concentrations can represent the number of viable, metabolically active cells, were formed by NADPH-dependent oxidoreductase enzymes contained in viable cells. The colored solution was quantified by measuring absorbance at 570/690 nm using a microplate reader (EZ Read 400, Biochrom, Holliston, MA, USA). The effect of H-HLA and L-HLA on DPSC cell viability was also tested over a period of 5 days.

#### 2.4.2. The Effect of L-HLA on the MAPK Signaling Pathways

To determine how the intracellular signaling cascade was affected by L-HLA in DPSCs, alternation in specific intracellular kinases in L-HLA-treated DPSCs were assessed. For the ERK, JNK, and p38 signal pathways, 50 µmol/L of PD98059 (Gibco/Invitrogen, Carlsbad, CA, USA), 30 µmol/L of SP600125 (Enzo Life Sciences, Farmingdale, NY, USA), and 10 µmol/L of SB203580 (Gibco/Invitrogen, Carlsbad, CA, USA) were added, respectively, to the culture medium. The cell viabilities were determined for cells treated with or without L-HLA. In addition, to ensure the binding of the prepared L-HLA to HLA receptor on DPSC membranes, MTT assay was performed after adding 0.4 μg/mL anti-CD44 antibody (Thermo Fisher Scientific, Waltham, MA, USA) to the cultured cell system, with and without L-HLA exposure.

### 2.5. In Vivo Animal Experiments

#### 2.5.1. HLA/CMC Preparation and Animal Model

In this study, HLA/CMC hydrogel was prepared prior to experiments by mixing 10 mg/mL CMC and HLA (30 mg/mL) in a 1:1 ratio. Six New Zealand White rabbits weighing 3.1–3.5 kg were used as test subjects. The rabbits were fed solid food and water and maintained in a clean and stable environment with a constant temperature of 25°C and humidity of 50%. Animal use and experimental protocols were designed according to the National Research Council’s Guide for the Care and Use of Laboratory Animals guidelines. The entire protocol was approved by the Institutional Animal Care and Use Committee of the National Defense Medical Center, Taipei, Taiwan (IACUC-18-245). After standard sterilization procedures, rabbits were anesthetized with an intramuscular injection of Zoletil 50 at a dose of 15 mg/kg (Virbac, Carros, France). After deep anesthesia was achieved, surgical sites were shaved and disinfected with povidone iodine (Sigma-Aldrich, St. Louis, MO, USA) (Figure 1b). The artificial bone defects were prepared according to previous studies assessing dental grafting materials [32,33]. Briefly, four critical circular defects (d = 6 mm) were prepared in the parietal bone (Figure 1b). To follow the 3R spirit of the Declaration of Helsinki and to eliminate artifacts due to experimental error caused by inter-individual differences, twelve defects on three rabbits were randomly assigned to prepared H-HLA/CMC, L-HLA/CMC, and HLA-free CMC (control) groups (n = 4). To reduce infection, rabbits received postoperative antibiotics and analgesics for 3 days via intramuscular injection. After 4 and 8 weeks of healing, the rabbits were euthanized under anesthesia (50 mg/mL Zoletil, dosage 15 mg/kg) by CO_2_ gas asphyxiation. Bone tissues were cut using a surgical burr. Immediately after bone samples were collected, 10% formaldehyde solution was used to fix and preserve the tissue. All animals completed the entire experiment. To ensure a blinded experiment, different operators prepared grafting material and filled artificial defects. The experiment report followed ARRIVE (Animal Research: Report of in vivo Experiments) guidelines. 

#### 2.5.2. Micro-CT Measurements

Micro-CT was used to observe bone growth conditions with various grafting materials. Collected bone tissues were scanned using a micro-CT scanner (SkyScan 1076, Bruker, Kontich, Belgium) with the following parameters: energy level: 75 kV, current: 200 μA, pixel size: 18 μA. After 3D images were reconstructed, new bone formation was calculated using analysis software (CTAn, Bruker). According to previous studies [22,23], new bone was defined as the ratio of bone volume percentage (BV) to total tissue volume (TV). In addition, trabecular thickness, trabecular number, and trabecular separation in the defect area were measured using the micro-CT images.

#### 2.5.3. Histological and Histomorphometrical Evaluation

For histological analysis, a decalcification procedure was performed on collected bone tissues by immersion in 10% EDTA (0.1 M phosphate buffer, pH 7.4) (Thermo Scientific, Waltham, MA, USA) for 4 weeks. Sample tissues were then dehydrated in alcohol with gradient concentrations from 60 to 100%. After embedding the samples in paraffin, 5 μm thick sections were prepared. To observe changes in bone growth in defects treated with different HLA conditions, sample sections were stained with hematoxylin and eosin (Sigma-Aldrich, St. Louis, MO, USA). Histological images were acquired and digitalized using a microscope slide scanner (OPTIKA, Ponteranica, Italy). The percentages of new bone, connective tissue, and remaining grafting materials were calculated using image analysis software (ImageJ, National Institutes of Health, Bethesda, MD, USA).

### 2.6. Statistical Analysis

Data are presented as mean values and standard deviations. Differences between samples were investigated using one-way analysis of variance (ANOVA) with Duncan’s post hoc test (SPSS Inc., Chicago, IL, USA). For all tests, *p* values less than 0.05 were considered statistically significant.

## 3. Results and Discussion

Due to sinus spaces, existing bone does not have the mechanical strength to achieve the ideal osseointegration with implants. Accordingly, maxillary sinus elevation surgery was developed to increase the amount of bone in preparation for dental implant placement [4,28,29]. It remains difficult, however, to perform this procedure in the posterior maxilla area due to graft material sagging. To address this issue, HLA was added to bone grafts to increase bone growth and improve handling [12,13,14]. The mechanism of HLA-induced osteo-regeneration varied according to the MW. For example, HLA with an MW ranging from 5 to 20 kDa exhibited bone regeneration induced by cytokine expression [22,34]. However, L-HLA with an MW of 30 and 60 kDa increased the osteogenic effect by promoting mesenchymal cell differentiation [35] and upregulating osteocalcin mRNA expression [36]. Osteo-regeneration of HLA with an MW ranging from 50 to 200 kDa is achieved by activating stem cells [21,24]. Alternatively, 390 kDa of HLA enhanced osteogenesis by increasing the cell adhesion rate and cell proliferation [21]. As the MW of the L-HLA used in this study was approximately 200 kDa, its effects on dental pulp stem cell viability were assessed. The characteristics of the DPSCs used in this study were identified using surface markers. Figure 2 shows that cells extracted from dental pulp (Figure 2a) positively expressed CD29, CD44, CD73, CD90, CD105, and STRO-1 (Figure 2d–i) and did not express hematopoietic markers CD14 and CD34 (Figure 2b,c). In addition, the cells’ osteogenic differentiation ability was confirmed by the presence of calcium deposition after being induced by the osteogenic medium (Figure 3) as in previous studies [37,38,39]. These characteristics meet the DPSC criteria as described in previous studies [30,40]. 

Previous studies have indicated that both H-HLA [10,11] and L-HLA [24,25,41] can promote bone growth. However, the amount of HLA needed to exhibit osteogenesis is dependent on the HLA MW [21]. For in vitro studies, L-HLA (30 and 40 kDa) with a concentration of 1.0 mg/mL has been reported to promote MSC osteogenesis [35], but the effective concentration increased to 4 mg/mL when H-HLA (800 kDa) was used [42]. The MTT assay, which is based on mitochondrial activity in cells, shows that the L-HLA increased cell proliferation at concentrations above 1.0 mg/mL after 24 h of culturing (Figure 4a). This result confirmed the finding that mouse mesenchymal stem cells exhibit bone colony formation in vitro with L-HLA (30 and 40 kDa) in 1.0 and 2.0 mg/mL concentrations [39]. Because DPSCs play an important role in guided bone regeneration surgery [43,44], the L-HLA used in this study exhibits potential for use as a bone grafting material. In addition, our result falls in agreement with a previous study by Zhao et al. which also indicated that the MW and HLA concentrations interacted to affect cell proliferation [21]. Because cell proliferation is an early event in the tissue regeneration process, it is reasonable to conclude, as did Zhao and colleagues, that L-HLA’s positive effect on bone growth is due to its bio-function at the early stage of the stem cell culture. Additionally, Figure 4a shows that exposure to L-HLA in higher amounts did not significantly increase proliferation. Because cell viability at concentrations higher than 1.0 mg/mL was not significantly higher, HLA concentrations of 1.0 mg/mL were used in all further experiments. The cell source is another factor that contributes to the different effect of HA on MS cells [20]. Figure 4b shows that DPSC proliferation increased during the culture period due to the addition of H-HLA and L-HLA after two days. However, there was no significant difference in viability between the H-HLA and L-HLA groups at a concentration of 1 mg/mL. This may be due to the stem cells used in this study having been extracted from dental pulp tissue, a different tissue than was used in the studies mentioned above. Because DPSCs play an important role in guided bone regeneration surgery [43,44], the L-HLA used in this study exhibits potential for use as a bone grafting material.

Mitogen-activated protein kinase (MAPK) regulates several cellular functions, such as proliferation, differentiation, motility, and survival [30]. MAPKs can be divided into three main pathways: extracellular signal-regulated kinases (ERK), Jun N-terminal kinases (JNK), and p38 mitogen-activated protein kinase (p38 MAPK). In this study, inhibitors were used to explore whether the effects of L-HLA on dental pulp stem cells are related to these pathways. Figure 5 shows the results of cell proliferation when DPSCs were cultured with different signaling cascade inhibitors. The addition of L-HLA significantly promoted the growth of DPSCs in all groups (Figure 4). However, the cell viability decreased when the ERK inhibitor (PD98059) (Figure 5a), p38 inhibitor (SB203580) (Figure 5b), and JNK inhibitor (SP600125) (Figure 5c) were added to the cultures. Because the L-HLA-induced cell viability was significantly reduced by the MAPK inhibitors, it is reasonable to suggest that MAPK signaling pathway activation is responsible for enhanced L-HLA-induced DPSC proliferation. It is well-known that the expression of various osteogenesis-related markers such as ALP, RUNX-2, and OCN can be affected by MAPK pathway regulation [45]. However, these markers may be down-regulated, by H-HLA in particular, at higher concentrations [46]. That is, the data shown in Figure 5 agree with the previous finding that the mechanism by which L-HLA and H-HLA promote bone growth is different [21].

To clarify the mechanism of L-HLA’s positive effect on MAPK signaling, DPSCs were co-cultured with L-HLA and anti-CD44, and the cell viability was measured. As shown in Figure 5d, the absorbance value without the L-HLA and anti-CD44 antibody was 0.313 ± 0.033, while adding L-HLA but not the anti-CD44 antibody increased the absorbance to 0.428 ± 0.032 (*p* < 0.01). However, when the anti-CD44 antibody was added, the absorbance decreased to 0.356 ± 0.014, a value significantly lower than seen in the group without the anti-CD44 antibody (*p* < 0.05). Because CD44 is a glycoprotein found on the surface of cell membranes and is a receptor for hyaluronic acid [47,48] and CD44 was expressed to a large degree on the tested DPSC cells (Figure 2e), this confirms that the L-HLA’s effect of promoting DPSC growth is mediated through CD44 (Figure 5d) as reported previously [49,50,51]. 

Figure 6 shows 3D images derived from a CT analysis of artificial bone defects at 4 and 8 weeks. CMC combined with either H-HLA (Figure 6b) or L-HLA (Figure 6c) increase bone growth compared to HLA-free samples at 4 weeks (Figure 6a). After 8 weeks of healing, obvious bone in-growth was seen at the defect boundaries even when HLA was not present (Figure 6d). For the H-HLA group, new bone formation was found at the central part of the defect at 8 weeks (Figure 6e). At the same time, newly formed bone covered almost the entire defect area for samples grafted with the L-HLA/CMC hybrid (Figure 6f). Although in vitro data showed that both H-HLA and L-HLA exhibit similar osteogenic potential (Figure 4), these in vivo experiments suggest that bone defects treated with a bone graft containing L-HLA exhibit better bone reparative processes compared to analogs with H-HLA. Quantitative analysis shows that new bone formation in bone defects treated with H-HLA and L-HLA complexes were 1.88 (*p* < 0.05) and 3.67 (*p* < 0.01) times higher than defects treated with pure CMC at 4 weeks (Figure 7a). These values become 1.87 (*p* < 0.01) and 2.95 (*p* < 0.01) times at 8 weeks. The bone formation in the L-HLA/CMC group is almost two times higher than in the defects filled with the H-HLA/CMC (*p* < 0.05). No difference in the trabecular thickness (Figure 7b), trabecular number (Figure 7c), or trabecular separation (Figure 7d) was observed when the H-HLA was used. Trabecular indications, however, were significantly improved when the defects were filled with bone grafts containing L-HLA, results which are consistent with a previous report that indicated L-HLA exhibits more osteoconductivity than H-HLA [25]. With these results, we confirm that the major role of H-HLA in graft material is maintaining a stable shape rather than directly affecting bone regeneration [16]. As cellular studies cannot show the long-term true condition in vivo, our results can explain the ambiguity of a high MW HLA in bone tissue engineering [18,52]. 

A similar finding that artificial bone defects filled with L-HLA result in better bone healing was obtained from the histological experiments (Figure 8). At 4 weeks, although newly formed bone (15.65 ± 5.81%) was found when the artificial defect was filled with the H-HLA/CMC bone graft (Figure 8a), there was no statistical difference when compared to the HLA-free grafting material (9.77 ± 5.66%). However, the quantification of the histological images showed significant increases in new bone formation (28.47 ± 6.09%, *p* < 0.01) in the defects treated with the L-HLA/CMC complex. At 8 weeks, this value increased to 73.90 ± 9.22%, significantly higher than the samples filled with either the HLA-free bone graft or the H-HLA groups (25.67 ± 3.10% and 47.49 ± 8.67%, respectively, *p* < 0.01). In addition, the amount of connective tissue was about 60% for all three groups at 4 weeks. At this time point, no differences were observed among the groups; however, these tissue values become 51.82 ± 7.49 and 25.16 ± 9.43% for the H-HLA/CMC and L-HLA/CMC groups, respectively, significantly lower (*p* < 0.05) than in the defect sample treated with the HLA-free CMC (73.38 ± 9.32%) (Figure 9b). After eight weeks of healing, no residual bone substitute was found in any of the three groups. These data, when taken in combination with the CT results (Figure 6 and Figure 7), lead to a reasonable conclusion that L-HLA provides a better osteo-regenerative effect on bone reparation compared to H-HLA. This finding can be used to modify Aguado et al.’s 2014 conclusion that suggested H-HLA serves as a vehicle only and does not affect bone remodeling [16]. In this study, we found that H-HLA indeed plays a role in the bone remodeling process when mixed with a bone graft as in previous reports [10,12,32,53]. However, the effect is more obvious when H-HLA is replaced with L-HLA. This conclusion is consistent with Kuo et al.’s report that used bone graft complexes with various HLA and L-HLA mixing ratios to test bone defect healing efficiency in the femoral condyle of rabbit legs. The authors also concluded that higher L-HLA ratios result in better new bone formation [25]. 

As mentioned above, H-HLA’s osteogenic effect was reported to be mechanically oriented. This is because H-HLA assists osteogenesis through viscosity to form an adhesive scaffold system. This effect can alter the pore size on the surface of graft materials [10,11], thereby enhancing cell adhesion to increase the ability of bone cells to proliferate and migrate [36]. Indeed, H-HLA acts mainly as a companion material that enhances bone growth in bone grafts [16]. Unlike H-HLA, which cannot directly bind to cell receptors, the L-HLA used in this study bound to CD44 on the stem cell surfaces (Figure 5d) and thus affected the MAPK signaling (Figure 5a–c). Therefore, L-HLA exhibits better bone growth performance when used as a component of the bone grafting material (Figure 6, Figure 7, Figure 8 and Figure 9). However, the disadvantage of using L-HLA as a component in combination with a bone graft is an inability to retain viscosity [21,25], making it difficult to handle at the posterior maxilla due to sagging. Although CMC worked to increase the L-HLA viscosity as in a previous study, using a bone graft composite containing an H-HLA/L-HLA hybrid as per Kuo et al.’s suggestion should be also recommended when applying this material in bone tissue engineering [25].

## 4. Conclusions

In this study, a new material made by an L-HLA/CMC hybrid was fabricated and tested using an animal model. We found that L-HLA exhibited a stronger osteo-regenerative effect compared to H-HLA. Although a decrease in the HLA MW results in the loss of the material’s viscosity, this can be compensated by adding CMC. Although a limitation exists in this study in that all the experiments were performed using a rat animal model that does not fully represent conditions in the human oral cavity, our data show that the L-HLA/CMC hybrid fabricated in this study provides better osteoconductive properties and significant intracellular responses for bone grafts, making it reasonable to suggest this L-HLA/CMC hybrid as a potential novel dental material for guided bone regeneration surgery.

## Figures and Tables

**Figure 1 polymers-14-03211-f001:**
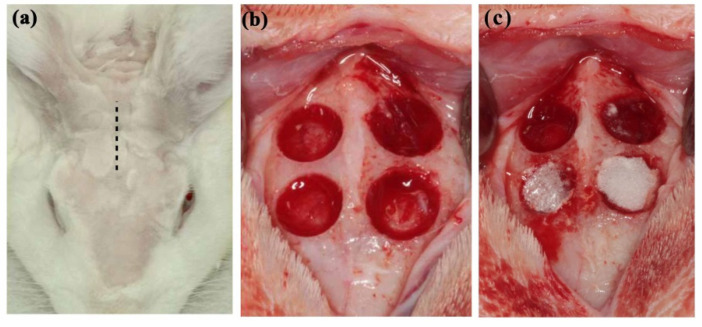
Animal model used in this study. (**a**) Before test, the surgical sites were shaved. (**b**) Four circular bone defects with a diameter of 6 mm were prepared on each rabbit skull. (**c**) Prepared defects were filled with HLA/CMC hybrids.

**Figure 2 polymers-14-03211-f002:**
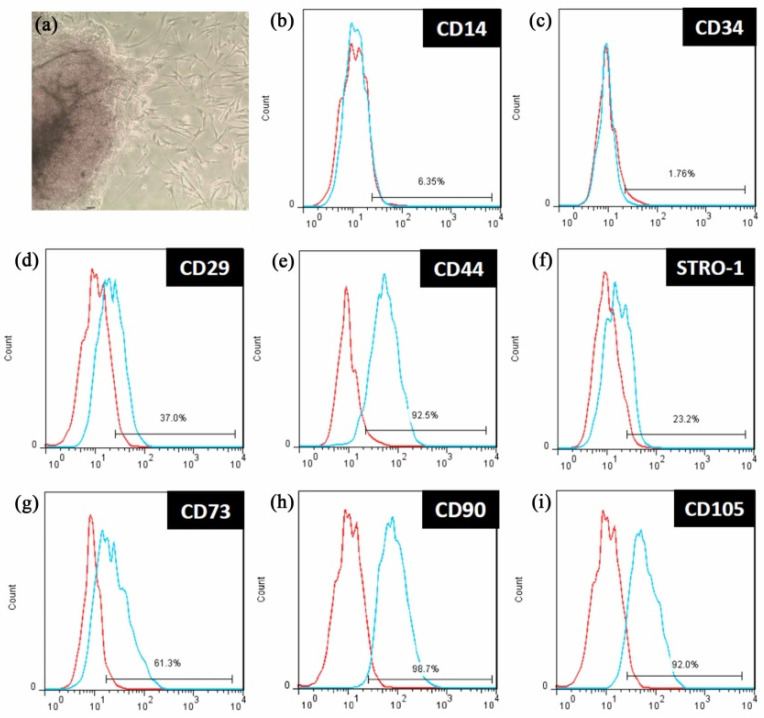
Characteristics of isolated dental pulp stem cells (**a**) used in this study were confirmed by negative expression of (**b**) CD14 and (**c**) CD34 but positive expression of (**d**) CD29, (**e**) CD44, (**f**) STRO-1, (**g**) CD73, (**h**) CD90, and (**i**) CD105. Blue and red lines represent the positive-stained and control cells, respectively.

**Figure 3 polymers-14-03211-f003:**
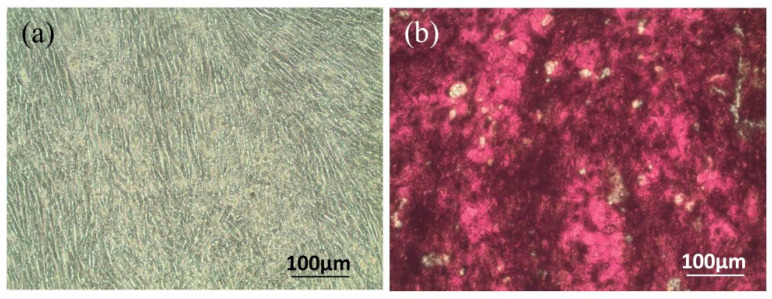
Osteogenic test of the dental pulp stem cells used in this study. (**a**) No mineralized nodule deposition was found in cells cultured with general medium. (**b**) In the osteogenic group, nodule-like structure depositions stained in red were observed throughout the cell layer.

**Figure 4 polymers-14-03211-f004:**
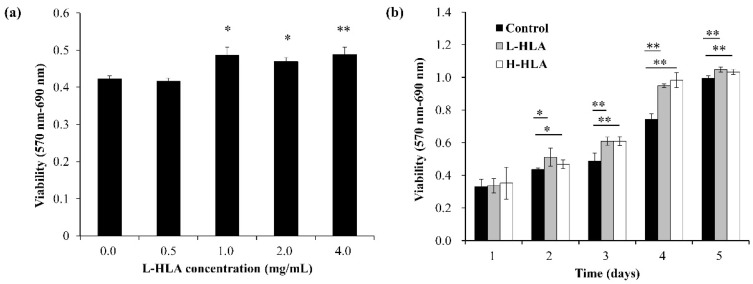
(**a**) L-HLA at concentrations above 1.0 mg/mL increased DPSC cell viability in serum-free medium. (**b**) DPSC cells treated with L-HLA and H-HLA for 5 days. Data are presented as mean ± standard error in at least three independent experiments. Significance is indicated as * (*p* < 0.05) and ** (*p* < 0.01).

**Figure 5 polymers-14-03211-f005:**
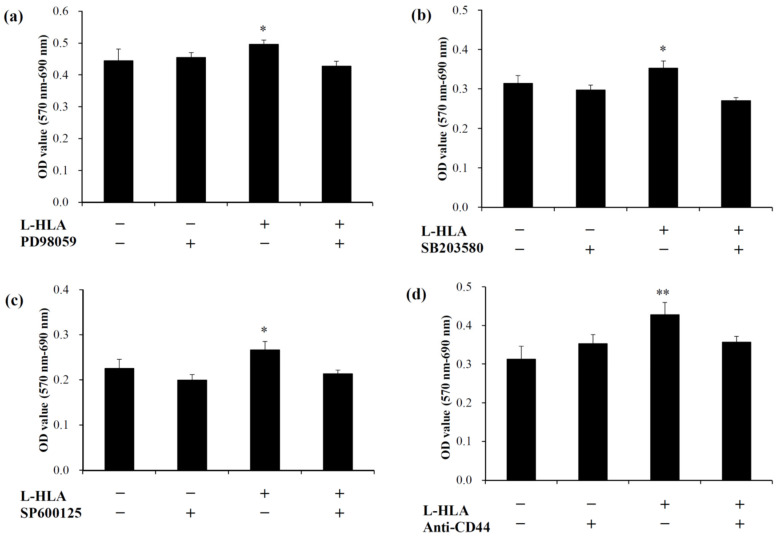
Growth of DPSCs cultured with L-HLA. L-HLA promoted DPSC growth; however, MAPK inhibitors such as (**a**) ERK inhibitor PD98059, (**b**) p38 inhibitor SB203580, (**c**) JNK inhibitor (SP600125) caused significantly lower viability when compared to cells treated without an inhibitor. The similar inhibited effect was found when the cells were treated with anti-CD44 (d). Significance is indicated as * (*p* < 0.05) and ** (*p* < 0.01).

**Figure 6 polymers-14-03211-f006:**
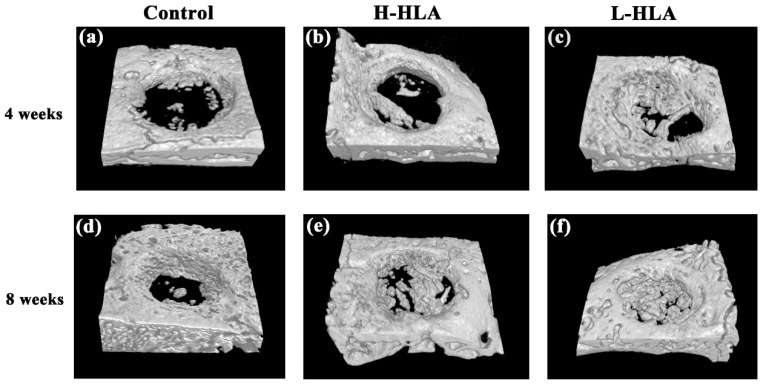
Micro-CT 3D images of tested bone defects grafted with HLA-free HA/β-TCP bone graft (**a**,**d**), bone graft mixed with H-HLA (**b**,**e**), and L-HLA-HA/β-TCP (**c**,**f**) at 4 weeks (**a**–**c**) and 8 weeks (**d**–**f**) post healing.

**Figure 7 polymers-14-03211-f007:**
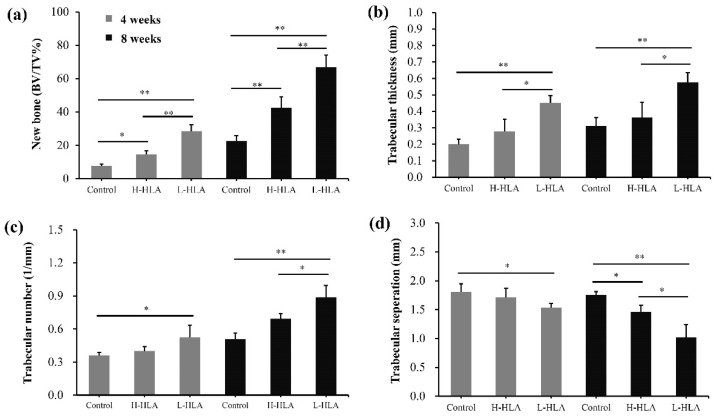
Micro-CT analysis of new bone formation (**a**), trabecular thickness (**b**), trabecular number (**c**), and amount of trabecular separation (**d**). Data are expressed as mean + standard deviation. Significance is indicated as * (*p* < 0.05) and ** (*p* < 0.01).

**Figure 8 polymers-14-03211-f008:**
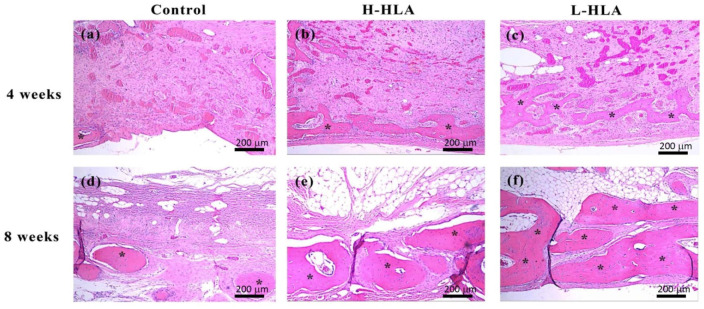
Histological images of bone tissue at artificial defects created in this study (magnification 200×). Microscopy shows HLA-free bone graft (**a**,**d**), bone graft mixed with H-HLA (**b**,**e**), and L-HLA-HA/β-TCP (**c**,**f**) at 4 (**a**–**c**) and 8 weeks (**d**–**f**) post healing (* denotes newly formed bone). Scale bar = 200 μm.

**Figure 9 polymers-14-03211-f009:**
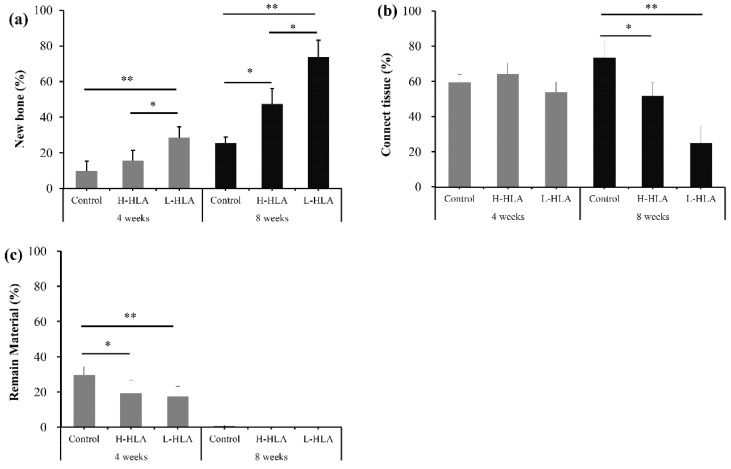
Analysis of measurements from histological images of (**a**) new bone formation, (**b**) connective tissue, and (**c**) remaining material. Data are expressed as mean + standard deviation. Significance is indicated as * (*p* < 0.05) and ** (*p* < 0.01).

## Data Availability

Not applicable.

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
