# Peer review of "Fabrication of Low-Molecular-Weight Hyaluronic Acid–Carboxymethyl Cellulose Hybrid to Promote Bone Growth in Guided Bone Regeneration Surgery: An Animal Study"

_polymers, 2022, doi:10.3390/polym14153211_

Round 1

Reviewer 1 Report

The scientific paper "Fabrication of Low Molecular Weight Hyaluronic Acid-Carboxymethyl Cellulose Hybrid to Promote Bone Regeneration” aimed to fabricate an L-HLA-carboxymethyl cellulose (CMC) hybrid to promote bone regeneration while maintaining viscosity. I can make the following considerations:

1)      The abstract needs to be revised. There are abbreviations that have not been properly clarified in relation to their content. For example: MAPK pathways should be changed to “mitogen-activated protein kinase (MAPK) pathway”. Likewise with ERK and JNK.

2)      The introduction is too short. I suggest creating a new paragraph, after the first one, about the types of graft (autogenous, allogeneic, xenogeneic, synthetic) and their advantages and disadvantages. I suggest entering some options that can facilitate the insertion of particulate grafts (biocomplexes) doi: 10.3390/polym14102075

3)      At the end of the manuscript, after funding, the Institutional Review Board Statement and Informed Consent Statement must be inserted (according to the instructions for authors Polymers MDPI).

4)      MDPI endorses the ARRIVE guidelines (arriveguidelines.org/) for reporting experiments using live animals. Did the authors follow the guidance?

5)      Is the 6 mm defect in rabbits critical or non-critical size? Please discuss this.

6)      Figure 1 is very poor. Please complement with the defects filled in with their respective grafts.

7)      Figure 1 demonstrates 4 defects but the authors report 3 groups as seen in the results: Control, H-HLA and L-HLA. Please clarify.

8)      Figure 8 must be changed by inserting the scale bar, and in the legend put the objective used.

9)      The results were well positioned but the discussion about these results was very brief. It can be noticed by the low number of references used. I suggest increasing the discussion.

10)  Complement the conclusions, with a brief beginning stating the objective of the study, and at the end its limitations and clinical applicability

Author Response

Comment 1.1: The abstract needs to be revised. There are abbreviations that have not been properly clarified in relation to their content. For example: MAPK pathways should be changed to “mitogen-activated protein kinase (MAPK) pathway”. Likewise with ERK and JNK.

Author Response: We sincerely thank the reviewer for pointing out these abbreviations. In the revised Abstract, these phrases were changed to Mitogen-activated protein kinase (MAPK), extracellular-signal-regulated kinase (ERK) and c-Jun N-terminal kinase (JNK).

Comment 1.2: The introduction is too short. I suggest creating a new paragraph, after the first one, about the types of graft (autogenous, allogeneic, xenogeneic, synthetic) and their advantages and disadvantages. I suggest entering some options that can facilitate the insertion of particulate grafts (biocomplexes) doi: 10.3390/polym14102075

Author Response: We thank the reviewer for this comment. In the revised manuscript, we added a new paragraph in Introduction to introduce the four types of grafts and the recent application of biocomplex.

Comment 1.3: At the end of the manuscript, after funding, the Institutional Review Board Statement and Informed Consent Statement must be inserted (according to the instructions for authors Polymers MDPI).

Author Response: We thank the reviewer for the insightful comments. “Institutional Review Board Statement” was added at the end of the manuscript after funding.

Comment 1.4: MDPI endorses the ARRIVE guidelines (arriveguidelines.org/) for reporting experiments using live animals. Did the authors follow the guidance?

Author Response: We thank the reviewer for pointing this out. After checking the ARRIVE guidelines, we revised the Material and Methods for reporting animal experiments according to the regulation of the guideline. The sample size, randomization, and blinding procedures were also added to the revised manuscript.

Comment 1.5: Is the 6 mm defect in rabbits critical or non-critical size? Please discuss this.
Author Response: According to a previous report, the 6-mm artificial bone defect in the rabbit model is a critical size and has been extensively used in implantation studies (Blanco et al. Stem Cells International, Volume 2018, Article ID 7089484).

Comment 1.6: Figure 1 is very poor. Please complement with the defects filled in with their respective grafts.

Author Response: We thank you for the reviewer’s suggestion. In the revised Figure 1, a new picture demonstrating the defects filled with HLA/CMC was added.

Comment 1.7: Figure 1 demonstrates 4 defects but the authors report 3 groups as seen in the results: Control, H-HLA, and L-HLA. Please clarify.

Author Response: Thank you for the reviewer’s comment. In the revised manuscript, the description of sample size and grafting situation was modified as follows:

“To follow the 3R spirit of the Declaration of Helsinki and to eliminate artifacts due to experimental error caused by inter-individual differences, twelve defects on three rabbits were randomly filled with prepared H-HLA/CMC, L-HLA/CMC, and HLA-free CMC (control group) (n = 4)”.

Comment 1.8: Figure 8 must be changed by inserting the scale bar, and in the legend put the objective used.

Author Response: We thank you for the reviewer’s suggestion. In the revised manuscript, scale bars were added in Fig. 8 as well as in the legend.

Comment 1.9: The results were well positioned but the discussion about these results was very brief. It can be noticed by the low number of references used. I suggest increasing the discussion.

Author Response: Thank you for the reviewer’s suggestion. In the revised manuscript, the discussions for cell experiments were increased after each paragraph of the results. A full paragraph was used to discuss the animal study at the last paragraph of the Discussion section. The number of reference was increased from 39 to 53.  

Comment 1.10: Complement the conclusions, with a brief beginning stating the objective of the study, and at the end its limitations and clinical applicability

Author Response: Thank you for the reviewer’s suggestion. In the revised manuscript, the conclusion was re-written with a brief beginning statement of the objective of the study. The limitations and clinical application of this study were also included.

Reviewer 2 Report

Overall, this study is novel and complete. Few minor comments: 

Minor Comments:

Consider to change the title appropriately based on the work done with dental and skull. Now the title did not reflect the work in a proper way.

The introduction part is not convincing, Specify the previous works on L-HLA and CMC in dental research.  Why this study has been undertaken and how does this study address the existing scientific gap and major proposal of this study with a brief outcome.

Try to elaborate the method section with more details procedures, especially characterization. 

Revise the conclusion part a bit more to elaborate on the major outcomes, novelty, conclusion, future perspectives and limitations of the study.

Author Response

Reviewer 2:

Overall, this study is novel and complete. Few minor comments:

Author Response: We thank this comment from the reviewer.

* Comment 2.1: Consider to change the title appropriately based on the work done with dental and skull. Now the title did not reflect the work in a proper way.

Author Response: We really appreciate the reviewer’s concerns and comments. In the revised manuscript, the title was changed to “Fabrication of Low Molecular Weight Hyaluronic Acid-Carboxymethyl Cellulose Hybrid to Promote Bone Growth in Guided Bone Regeneration Surgery: An Animal Study”.

* Comment 2.2: The introduction part is not convincing, Specify the previous works on L-HLA and CMC in dental research. Why this study has been undertaken and how does this study address the existing scientific gap and major proposal of this study with a brief outcome.

Author Response: We really thank for this comment from the reviewer to modify our manuscript. In the revised manuscript, we added more descriptions to introduce the dental research of HLA/CMC and the importance of the current study in the second and last paragraphs of the revised manuscript.

* Comment 2.3: Try to elaborate the method section with more details procedures, especially characterization.

Author Response: We thank you for the reviewer’s suggestion. The characterization procedure of DPSCs was re-written to provide detailed information of the method. In addition, a more detailed description of the animal study was added to the section for providing the grouping method and ethical issue.

* Comment 2.4: Revise the conclusion part a bit more to elaborate on the major outcomes, novelty, conclusion, future perspectives and limitations of the study.

Author Response: We really appreciate the reviewer’s concerns and comments. We thank you for the reviewer’s suggestion. In the revised manuscript, the conclusion was re-written with a longer description of major outcomes, novelty, conclusion, future perspectives, and limitations of the study.

Reviewer 3 Report

Dear Authors,

The submitted manuscript succeeds in filling the knowledge gap on the effect of L-HLA/CMC bone graft material on bone defect healing, as initially stated. The complex biological evaluation is an important asset, along with the provided Discussion part, very on-point. 

Author Response

Reviewer 3:

The submitted manuscript succeeds in filling the knowledge gap on the effect of L-HLA/CMC bone graft material on bone defect healing, as initially stated. The complex biological evaluation is an important asset, along with the provided Discussion part, very on-point. 

Author Response: We really thank for the reviewer’s positive feedback on our manuscript.

Round 2

Reviewer 1 Report

No comments